# Large Language Models and Genomics for Summarizing the Role of microRNA in Regulating mRNA Expression

**DOI:** 10.3390/biomedicines12071535

**Published:** 2024-07-10

**Authors:** Balu Bhasuran, Sharanya Manoharan, Oviya Ramalakshmi Iyyappan, Gurusamy Murugesan, Archana Prabahar, Kalpana Raja

**Affiliations:** 1School of Information, Florida State University, Tallahassee, FL 32306, USA; bbhasuran@fsu.edu; 2Department of Bioinformatics, Stella Maris College, Chennai 600086, Tamil Nadu, India; sharanya.bioinfo@gmail.com; 3Department of Computer Science and Engineering, Amrita School of Computing, Amrita Vishwa Vidyapeetham, Chennai 641112, Tamil Nadu, India; iroviya@gmail.com; 4Department of Computer Science and Engineering, Koneru Lakshmaiah Education Foundation, Green Fields, Guntur District, Vaddeswaram 522302, Andhra Pradesh, India; gurusamy.m@hotmail.com; 5Center for Gene Regulation in Health and Disease, Department of Biological, Geological, and Environmental Sciences (BGES), Cleveland State University, Cleveland, OH 44115, USA; a.prabahar@csuohio.edu; 6Department of Biomedical Informatics and Data Science, School of Medicine, Yale University, New Haven, CT 06510, USA

**Keywords:** machine learning, deep learning, large language models, natural language processing, genomics, miRNA–mRNA interactions

## Abstract

microRNA (miRNA)–messenger RNA (mRNA or gene) interactions are pivotal in various biological processes, including the regulation of gene expression, cellular differentiation, proliferation, apoptosis, and development, as well as the maintenance of cellular homeostasis and pathogenesis of numerous diseases, such as cancer, cardiovascular diseases, neurological disorders, and metabolic conditions. Understanding the mechanisms of miRNA–mRNA interactions can provide insights into disease mechanisms and potential therapeutic targets. However, extracting these interactions efficiently from a huge collection of published articles in PubMed is challenging. In the current study, we annotated a miRNA–mRNA Interaction Corpus (MMIC) and used it for evaluating the performance of a variety of machine learning (ML) models, deep learning-based transformer (DLT) models, and large language models (LLMs) in extracting the miRNA–mRNA interactions mentioned in PubMed. We used the genomics approaches for validating the extracted miRNA–mRNA interactions. Among the ML, DLT, and LLM models, PubMedBERT showed the highest precision, recall, and F-score, with all equal to 0.783. Among the LLM models, the performance of Llama-2 is better when compared to others. Llama 2 achieved 0.56 precision, 0.86 recall, and 0.68 F-score in a zero-shot experiment and 0.56 precision, 0.87 recall, and 0.68 F-score in a three-shot experiment. Our study shows that Llama 2 achieves better recall than ML and DLT models and leaves space for further improvement in terms of precision and F-score.

## 1. Introduction

MicroRNAs (miRNAs), a class of non-coding RNAs (ncRNAs), are single-stranded and contain approximately 18–26 nucleotides. miRNAs exert the post-transcriptional regulation of gene expression by binding to microRNA-responsive elements (mREs) on target mRNAs (or the genes). The canonical biogenesis of miRNAs involves four sequential steps: (i) the transcription of pri-miRNA by RNA polymerase II, forming a hairpin-like structure in the nucleus; (ii) cleavage by the microprocessor complex, comprising Drosha, an RNase III enzyme, and its cofactor DiGeorge Syndrome Critical Region 8 (DGCR8), generating pre-miRNA; (iii) exporting pre-miRNA to the cytoplasm via Exportin 5 and Ran Guanosine Triphosphate (Ran-GTP); and (iv) processing by the DICER complex to yield miRNA–miRNA* duplexes. The guide strand of mature miRNA, preferentially incorporated into the RNA-induced silencing complex (RISC), mediates target mRNA recognition through base pairing with complementary sequences in the 3′ untranslated region (UTR), leading to translational repression or mRNA degradation [1,2,3]. The guide strand of miRNA is more stable and functionally active in mediating miRNA-mediated gene regulation when compared to the passenger strand (miRNA* or miRNA-3p). Non-canonical biogenesis pathways, such as Drosha-independent (e.g., mirtron) and Dicer-independent (e.g., Argonaute-dependent) mechanisms, contribute to the production of additional miRNAs. Notably, a single miRNA can regulate multiple mRNAs, and conversely, a single mRNA can be targeted by multiple miRNAs [4]. miRNAs have been observed in various cellular compartments, including the nucleus, cytoplasm, mitochondria, and exosomes, each with distinct functional roles. These roles encompass transcriptional regulation, alternative splicing, metabolism, development, apoptosis, and intercellular communication [5]. Moreover, miRNAs are pivotal in regulating neuronal gene expression, brain morphogenesis, muscle differentiation, and stem cell division. Therefore, several studies aim to reveal their involvement in the development of various diseases, spanning cancer, cardiovascular disorders, inflammatory conditions, neurodevelopmental anomalies, and autoimmune disorders, as well as liver, skeletal muscle, and skin ailments [6]. The miRNA–mRNA Interaction Corpus (MMIC) and scripts developed for the current study are available at https://github.com/balubhasuran/miRNA_mRNA-Relation-Extraction (accessed on 2 July 2024).

Studies on miRNA and mRNA interactions have accumulated a huge volume of published papers over the years. Extracting these interactions manually is impossible. Alternatively, relation extraction, a popular task within natural language processing (NLP) can be applied to extract the miRNA and mRNA interactions automatically. Relation extraction is an established area within NLP and numerous works are available for extracting the relations between two or more biomedical entities from the articles in PubMed. While most of the approaches are specific to a pair of specific entities, i.e., protein–protein interaction [7], only a few approaches are capable of extracting the relations between any pair of biomedical entities existing in PubMed [8]. In addition, there are certain specific case studies that use PubMed to explore the association between specific entities such as T-2 Toxin, Cerebral Edema, and Aquaporin-4 [9]. 

Earlier approaches for relation extraction were rule-based systems [7]. Over the years, NLP approaches including relation extraction have evolved to include machine learning algorithms, deep learning algorithms [10,11], and transformer-based approaches [10,12,13]. Recently, large language models (LLMs) have gained huge interest among NLP researchers exploring their application in various NLP tasks including relation extraction [14,15]. In a recent study, we explored the performance of two LLM models, namely GPT-3.5-turbo and GPT-4, in extracting gene–disease association information from two standard corpora, namely EU-ADR [16,17] and the Gene Associations Database (GAD) [18], and chemical–protein interaction from ChemProt, a corpus released by the BioCreative VI shared task [19]. Our attempts showed that the performance of GPT is slightly lower than the transformer models such as BioBERT and PubMedBERT [20]. 

Several studies focused on extracting miRNA–mRNA interactions from PubMed. Li et al. introduced miRTex, a relation extraction system for extracting miRNA-target, miRNA-gene, and gene-miRNA regulations from 150 PubMed abstracts. miRTex achieved F-scores ranging from 0.88 to 0.94 for extracting different types of regulations [21]. Naeem et al. [22] developed miRSel, a NLP-based system for extracting miRNA–mRNA interactions. miRSel achieved an F-score of 0.76 for extracting miRNA-gene relations related to humans from 1973 sentences from PubMed [22]. In addition to relation extraction, miRSel also integrates microRNA, gene, and protein occurrences with existing databases by employing various dictionaries. Lamurias et al. [23] developed IBRel, a distant supervision-based microRNA-gene relation extraction system using a multi-instance learning approach. The system reported an F-score 28.3% higher on the IBRel-miRNA dataset developed using 318 relations. This is higher than the baseline evaluated on the same dataset. IBRel also demonstrated robustness using 27 miRNA-gene relations from cystic fibrosis [23]. Lou et al. developed a microRNA-target interaction extraction system by combining PubMedBERT and SciBERT with LSTM. The system was evaluated on miRTarBase data and reported to achieve F-scores of 0.845 for Sentence-Level Encoding (SLE), 0.800 for Concatenation of Word-Level Encoding (CWLE), and 0.801 for Average of Word-Level Encoding (AWLE) [24]. These example studies highlight the advancements and varying approaches in miRNA–mRNA relation extraction.

In recent times, LLMs have gained much attention among NLP researchers. LLMs are huge deep learning models that are pre-trained on vast amounts of data. LLMs are artificial intelligence (AI) systems capable of understanding and generating human language in various formats such as text, voice, and image. LLMs have moved NLP research to much more advanced research areas such as natural language understanding (NLU) and natural language generation (NLG). The Generative Pre-trained Transformer (GPT) released by Open AI opened the doors to exploring and developing many LLMs in various domains including biomedicine. Open AI’s recent models, GPT-3.5-turbo and GPT-4, moved NLP researchers to a new space for exploring all core tasks, including relation extraction. In addition to GPT, other LLMs such as Llama [25], Mistral [26], and Mixtral [27] allow researchers to generate models for various clinical and biomedical tasks. In the current work, we explored OpenAI’s GPT-3.5-turbo and GPT-4, Meta’s LLaMA-2, and Anthropic’s Claude-2 [28] models for extracting the interactions between miRNAs and mRNAs in PubMed abstracts. While GPT-3.5-turbo, GPT-4, and Claude-2 are proprietary models, Llama-2 is an open-sourced model. Currently, GPT-3.5-turbo, LLaMA-2, and Claude-2 are freely accessible. GPT-4 is a subscription-based LLM. 

In addition to extracting miRNA–mRNA interactions using NLP/LLMs, we applied the standard genomics approaches to understand the pathways related to miRNA and mRNA. Elucidating the pathways associated with miRNA and mRNA interactions provides insights into the broader biological context in which the regulatory processes occur [29]. Understanding these pathways allows us to decipher how miRNAs influence gene expression within specific biological pathways or networks. This knowledge aids in identifying key regulatory genes, predicting potential miRNA targets, and uncovering the functional implications of miRNA-mediated gene regulation in various biological processes and diseases. Several omics studies related to miRNA gene interactions are performed using similar approaches [30]. In addition, pathway-informed text mining with LLMs facilitates hypothesis generation by identifying novel associations or regulatory patterns between miRNAs, mRNAs, and pathways. These insights can guide further experimental investigations and contribute to the discovery of novel regulatory mechanisms or therapeutic targets.

In the current study, we use four LLMs, namely GPT-3.5, GPT-4, LLaMA-2, and Claude-2, to extract the miRNA–mRNA interactions from PubMed abstracts. We apply genomics approaches to summarize the role of miRNA in regulating gene expression. The major contributions include the following:The release of an annotated corpus called the miRNA–mRNA Interaction Corpus (MMIC).The extraction of the miRNA–mRNA interactions from PubMed abstracts using LLMs. To our knowledge, this is the first study to use LLM for extracting miRNA–mRNA interactions.Applying genomics to identify the pathways associated with miRNA and mRNA and to predict the novel associations between miRNAs and mRNA. Our approach uncovered many previously undiscovered regulatory mechanisms.

Our pipeline is expected to provide meaningful insights into miRNA–mRNA interactions from PubMed abstracts.

## 2. Materials and Methods

### 2.1. Overall Workflow 

Our NLP/LLM pipeline includes the following: (i) retrieval and preprocessing of related abstracts from PubMed using NLP, (ii) annotation of miRNA, mRNA, and their relations, and (iii) model generation using ML, DLT, and LLM. In addition, we included a standard genomics-based analysis. Figure 1 shows the workflow.

### 2.2. Retrieval and Preprocessing

We retrieved PubMed IDs (PMIDs) related to five chronic diseases namely chronic obstructive pulmonary disease (COPD), Alzheimer’s disease (AD), stroke, type 2 diabetes mellitus (T2DM), and chronic liver disease. We extracted the titles and abstracts related to the retrieved PMIDs. We excluded those without abstracts. We segmented the titles and abstracts into individual sentences and assigned the respective PMID. We used scispaCy, (https://spacy.io/universe/project/scispacy, accessed on 2 July 2024), a named entity recognition tool, and a curated list of miRNA [21] and mRNA [31] names to identify the miRNA and mRNA entities mentioned in the sentences.

We obtained the list of genes (i.e., gene ID) mentioned in each PubMed abstract by mapping the PubMed ID (PMID) to gene2pubmed (https://ftp.ncbi.nlm.nih.gov/gene/DATA/gene2pubmed.gz, accessed on 2 July 2024), a standard resource from the National Center for Biotechnology Information (NCBI). We filtered the genes related to humans by referring to taxon (i.e., 9606 for humans) in gene2pubmed. We obtained the gene symbol, aliases, and name by mapping the gene ID to Entrez Gene. We used the gene symbol, aliases, and name to filter the sentences with mRNA mentions related to humans.

### 2.3. Entity Recognition

We used the scispaCy tool along with a curated keyword list of mRNA and miRNA names to recognize the mRNA and miRNA mentions in the input sentences. scispaCy provides various machine learning NER models such as en_ner_craft_md, en_ner_bionlp13cg_md, and en_ner_jnlpba_md. 

### 2.4. Annotation 

Two annotators with domain expertise in genomics labeled 200 sentences for initial annotation. The annotators achieved a high percentage agreement of 88.2%. This indicates a substantial overlap in the labeling decisions made by the annotators across the dataset. The inter-annotator reliability, measured using Cohen’s Kappa coefficient, was 0.802. The value suggests almost perfect agreement beyond chance, which is statistically significant (i.e., *p*-value = 0). This high level of agreement, underscored by both percentage and Cohen’s Kappa, indicates that the annotations were consistent and reliable. Following the initial annotation and evaluation, the annotators labeled 1000 randomly selected sentences to obtain the MMIC corpus. The corpus includes two labels: positive, for sentences conveying an interaction between an miRNA and mRNA pair, and negative, for sentences not conveying an interaction between an miRNA and mRNA pair. Appendix A provides the annotation guidelines used by the annotators for annotating the sentences with miRNA and mRNA. The annotation pipeline is depicted in Figure 2.

### 2.5. Model Generation 

We generated three different classes of AI models using ML, DLT, and LLMs. For ML, we generated seven models using support vector machines (SVMs), logistic regression, K-nearest neighbors (KNNs), decision tree, random forest, extreme gradient boosting (XGBoost), and LightGBM. For DLT, we generated three models using ClinicalBERT, BioBERT, and PubMedBERT. For LLM, we generated three models based on a zero-shot experiment and three models based on three-shot experiments. These models were generated on top of the existing GPT-3.5, Claude 2, and Llama 2. For model generation, we randomly split the sentences from MMIC corpus into two portions: 700 sentences for training and 300 for testing. We ensured an equitable distribution of positive and negative sentences in both training and test datasets. 

#### 2.5.1. ML Models

We used seven ML algorithms, namely SVM, logistic regression, K-Nearest Neighbors (KNNs), decision tree, random forest, Extreme Gradient Boosting (XGBoost) and LightGBM, for generating ML models. SVMs are utilized for their robust classification capabilities, particularly in high-dimensional spaces, in their finding of the optimal hyperplane that separates different classes. Logistic regression, by contrast, uses the probabilities for classification problems with two possible outcomes and is effective for its simplicity and efficiency in cases of linear separability. KNN works on the principle that similar instances lie in proximity within the feature space; thus, classification is performed by a majority vote of an instance’s neighbors. Decision trees segregate the data into branches to make predictions, providing intuitive decision rules and ease of interpretation. Random forest, an ensemble of decision trees, offers improved accuracy through bagging and feature randomness, reducing the risk of overfitting inherent to individual decision trees. XGBoost is an advanced implementation of gradient-boosted decision trees designed for speed and performance, which has been successful in numerous ML competitions. LightGBM is another gradient-boosting framework that uses tree-based learning algorithms, optimized for distributed and efficient training, particularly on large datasets. Each of these models brings unique strengths and can be chosen based on the specific nature of the dataset, the complexity of the problem, and the computational efficiency required. All the models were trained in their default setting from the scikit-learn (version 1.4.2) python package. Many existing studies do use default settings for generating ML models using the algorithms mentioned above [32,33,34,35].

#### 2.5.2. DLT Models

DLT models have significantly advanced NLP tasks by introducing domain-specific language models like ClinicalBERT, BioBERT, and PubMedBERT. These models are variations of the BERT (Bidirectional Encoder Representations from Transformers) architecture. ClinicalBERT is tailored for clinical notes and electronic health records (EHRs). BioBERT is trained on a massive dataset including PubMed abstracts and PMC full-text articles. PubMedBERT is exclusively pre-trained from scratch on PubMed, ensuring its proficiency in understanding and processing the sophisticated language used in biomedical literature. These models serve as powerful foundations for generating predictive models. We used the model versions Dmis Lab biobert-base-cased-v1.2 for BioBERT, emilyalsentzer Bio_ClinicalBERT as ClinicalBERT, and Microsoft BiomedNLP-PubMedBERT-base-uncased-abstract-fulltext as PubMedBERT models. All models were downloaded from the Hugging Face Repository (https://huggingface.co/dmis-lab/biobert-v1.1, accessed on 2 July 2024; https://huggingface.co/emilyalsentzer/Bio_ClinicalBERT, accessed on 2 July 2024; https://huggingface.co/microsoft/BiomedNLP-BiomedBERT-base-uncased-abstract, accessed on 2 July 2024. The models can be accessed with a free account on Hugging Face Repository.

We fine-tuned the DLT models using PyTorch Build (2.2.2) with Python 3.8 and NVIDIA Compute Unified Device Architecture (CUDA) 11.8 version. For fine-tuning, the ‘batch size’ was set to 16. The ‘AdamW optimizer’ was employed with a learning rate of 8 × 10^−7^ to adjust the model weights gradually. The ‘epsilon’ parameter was set to 1 × 10^−8^ to enhance numerical stability during optimization. The training was conducted over 30 ‘epochs’, allowing the model with sufficient iterations to learn from the dataset while mitigating the risk of overfitting through exposure to the data. The ‘max_length’ parameter truncates or pads the input sequences to a fixed length of 128 tokens, ensuring uniformity in input size and optimizing computational efficiency. The addition of special tokens, attention masks, and the application of truncation are essential for the model to correctly interpret the start and end of sentences and manage varying sentence lengths; the padding strategy is set to the longest sequence in a batch, which standardizes input length without unnecessary computation on padding tokens. These parameters are critical for the models to effectively learn from the data and are chosen to optimize performance given the computational constraints and the nature of the specific downstream task.

#### 2.5.3. LLMs

We developed a detailed prompt (Figure 3) that serves as an input to the LLMs used in the study. We used two distinct learning paradigms: zero-shot learning, where the models are tasked to infer the relations without any prior specific examples, and three-shot learning, where the models are provided with three illustrative instances—encompassing both positive and negative relational sentences—to guide their performance. The approach evaluates the models’ abilities to predict the relations between miRNA and mRNA in the input sentence based on the limited information.

### 2.6. Data Visualization

We mapped the disease(s) related to each miRNA–mRNA pair annotated as positive in the MMIC corpus. We used Gephi (version 0.10.0), an open-source network analysis and visualization software, for generating the miRNA–mRNA–disease network. Our objective was to visualize the miRNA–mRNA pair across five chronic diseases, namely COPD, AD, stroke, T2DM, and chronic liver disease.

### 2.7. Pathway Enrichment Analysis

We performed a pathway enrichment analysis for mRNA and miRNA mentioned in the MMIC corpus. For mRNA, we used ReactomePA package v 1.38.0 [36]. For miRNA, we used miRPathDB v 2.0 [37]. ReactomePA utilizes a specific hypothesis test, such as a hypergeometric test or Fisher’s exact test, to retrieve the pathways associated with mRNAs. We used the hypergeometric test to determine if the number of given genes (differentially expressed genes (DEGs)) in a pathway was greater than what would be expected by chance. The *p*-value was calculated using the hypergeometric distribution (Equation (1)):(1)$$P(X=x)=\frac{\left(\begin{array}{c} M\\ x \end{array}\right)\left(\begin{array}{c} N-M\\ K-x \end{array}\right)}{\left(\begin{array}{l} N\\ K \end{array}\right)}$$

Here, N represents the total number of genes in the background set, M represents the total number of genes in the pathway, K represents the total number of DEGs, and x represents the number of DEGs in the pathway.

The calculated *p*-value represents the probability of observing x or more DEGs in the pathway by chance. Since multiple pathways were tested simultaneously, the *p*-values were adjusted to control the false discovery rate (FDR). This adjustment was carried out using standard methods such as the Benjamini–Hochberg procedure. The pathways with significant q-values (i.e., adjusted *p*-values < 0.05) associated with mRNA in ReactomePA were retrieved. The pathways associated with miRNA and its target mRNA were retrieved from miRPathDB [36]. To create an interactive heatmap, miRPathDB utilizes all significantly enriched pathways for miRNA targets, constructs a matrix of −log10-transformed *p*-values, and clusters similar miRNAs and pathways using hierarchical clustering.

## 3. Results

### 3.1. MMIC Corpus

The MMIC corpus from 390 PubMed abstracts (342 PubMed abstracts in the training set and 215 PubMed abstracts in the test set) include the annotations for miRNA–mRNA interactions for five chronic diseases, namely COPD, AD, stroke, T2DM, and chronic liver disease. The corpus includes 1000 annotated sentences, and it is meticulously balanced between positive and negative sentences. The training dataset includes 700 sentences, with 354 positives and 346 negatives, reflecting a nearly equal distribution. The test subset consists of 300 sentences, with 146 positives and 154 negatives (Table 1). The training dataset contains a total of 182 unique miRNA entities and 339 unique mRNA entities. In comparison, the test dataset includes 130 unique miRNA entities and 201 unique mRNA entities. This distinction highlights the diversity and quantity of miRNA and mRNA entities present in each dataset and provides a comprehensive overview of the molecular components analyzed within the datasets. We kept this symmetrical allocation for the development of robust ML models as it prevents the bias that could result from the overrepresentation of any category. Additionally, the equitable distribution across the training and test datasets allows the robust evaluation of model accuracy in varied real-world situations where the prevalence of outcomes may not be inherently skewed. The average sentence length is provided as a density plot in Figure 4.

### 3.2. Performance of ML, DLT, and LLM Models

Table 2 shows the performance of the ML, DLT, and LLM models using MMIC corpus. Among the ML models, XGBoost shows the best performance of 0.67 precision, 0.67 recall, and 0.66 F-score. Among the DLT models, PubMedBERT shows the best performance of 0.783 precision, 0.783 recall, and 0.783 F-score. Among the LLMs, Llama 2, in a zero-shot experiment, shows the best performance of 0.681 F-score. Interestingly, Claude 2 achieved the highest precision of 0.587 and Llama 2, in the three-shot experiement, achieved the highest recall of 0.87 among the LLMs. Among all the models, PubMedBERT achieved the best performance. The F1-score of the DLT models on test data for 30 epochs is given in Figure 5a–c. Details of the training and validation loss for PubMedBERT, BioBERT, and ClinicalBERT for 30 epochs are provided in Appendix A. Overall, the ML models generally exhibited a consistent and balanced performance (Appendix A). The DLT models, especially those fine-tuned on biomedical texts like BioBERT and PubMedBERT, showed the highest performance among all three types of models. The LLMs demonstrated varying performances, with notable strengths in specific areas like recall for Llama-2.

### 3.3. miRNA–mRNA–Disease Network

Apart from the five chronic diseases considered for retrieving the PubMed abstracts, the miRNA–mRNA relations are linked with other chronic diseases such as breast cancer, prostate cancer, colorectal cancer, and gastric cancer. miRNA–mRNA–disease network is provided in Appendix A.

### 3.4. Enrichment Analysis

An elucidation of the biological pathways associated with mRNA-miRNA relations was conducted through pathway enrichment analysis using the ReactomePA package on a curated list of mRNA. This package utilizes the *p*-value to find the significantly enriched pathways. The pathways with adjusted *p*-values (often referred to as q-values) below a certain threshold (i.e., 0.05) were considered to be significantly enriched. The pathways with a q-value < 0.05 were considered to be significantly enriched with DEGs, suggesting that they may be biologically relevant to the condition under study. The pathways with a q-value ≥ 0.05 were not considered to be significantly enriched. These pathways imply that any observed enrichment could be due to random chance. ReactomePA identifies the pathways based on q-value. The selected pathways were likely to be involved in the biological processes related to the diseases under study. This is based on the enrichment of DEGs.

We determined several pathways that are significantly associated with our mRNA list. These pathways provide insights into the underlying disease mechanisms. We filtered the significantly enriched pathways with a q-value threshold of < 0.05. The pathways related to cancer, apoptosis, and cellular stress responses were prominent among the significant findings (Figure 6). Our analysis shows that several identified pathways (e.g., PI3K-Akt signaling pathway, Interleukin signaling pathway) are directly implicated in cancer development and progression. The PI3K-Akt signaling pathway plays a crucial role in cell survival and growth, and the Interleukin signaling pathway is associated with cell proliferation and differentiation.

To further explore the functional implications of miRNAs and their target mRNAs, we utilized miRPathDB for pathway analysis. This approach allowed us to identify the pathways specifically associated with miRNA regulation and their potential impact on gene expression. The results from miRPathDB were visualized as a heatmap, facilitating a comprehensive view of the miRNA-regulated pathways and their target mRNAs. The heatmap revealed several key pathways regulated by miRNAs, including those involved in cancer, apoptosis, and signaling pathways, as illustrated in Figure 7.

The pathway enrichment analyses using ReactomePA and miRPathDB collectively highlighted the multifaceted roles of mRNA-miRNA interactions in disease mechanisms. The identification of significant pathways related to cancer, apoptosis, immune regulation, and metabolic processes underscores the complexity and importance of miRNA-mediated gene regulation. These findings provide a foundation for further experimental validation and offer potential therapeutic targets for disease intervention.

## 4. Discussion

Comparing our NLP/LLM pipeline with the existing approaches on extracting miRNA–mRNA interactions from PubMed abstracts is impossible for various reasons. Among the four existing approaches, miRTex [30] and miRSel [31] have been released as web-based tools. The dataset used for evaluating miRTex has been released as a structured text that includes PMID, miRNA, mRNA, direction, and relation type. However, the exact sentence with miRNA and mRNA from the PubMed abstract is not available. For a fair comparison with our pipeline, we need the exact dataset with annotation used by miRTex. The dataset used for evaluating miRSel is not available. Thus, evaluating our approach on the dataset used for miRSel is impossible. The dataset used for evaluating IBRel [32] and the approach developed by Lou et al. [33] are available as free-text without entity-level annotation for miRNA and mRNA. It is impossible to evaluate our approach on this dataset without entity-level annotation. We also explored the possibility of evaluating the existing approaches on our MMIC corpus. The web interface of both miRTex and miRSel are not functioning. Though the source code is available for IBRel and the approach developed by Lou et al., it is impossible to validate their performance on an MMIC corpus that includes annotations for miRNA and mRNA in the input sentences. 

Although LLMs are known to excel in tasks such as summarization and reasoning, there are reports indicating that LLMs underperform in information extraction (IE) tasks, particularly in biomedical domains such as relation extraction [29]. Our recent work on extracting relations using three standard corpora validates the underperformance of GPT-3.5.turbo and GPT-4 [29]. Interestingly, the current study on extracting miRNA–mRNA interactions from PubMed abstracts also shows the underperformance of LLMs when compared to PubMedBERT. 

ML models like SVM and Logistic regression show moderate effectiveness, with performance peaking around 0.65. These traditional models remain relevant, particularly in scenarios where computational efficiency is critical. In contrast, KNN underperforms within this group, likely due to its sensitivity to the choice of ‘k’ and the distance metric used in higher-dimensional spaces. Decision trees and their ensemble version, Random Forest, demonstrate better performance, with Random Forest slightly outperforming due to its ability to reduce overfitting by averaging multiple decision trees. Boosted ML models such as XGBoost and LightGBM exhibit improvements over traditional algorithms, highlighting the effectiveness of boosting techniques in managing bias–variance trade-offs. XGBoost leads in this category, showcasing the strength of gradient-boosting frameworks in achieving higher accuracy by sequentially correcting the errors of weak learners.

DLT models, including ClinicalBERT, BioBERT, and PubMedBERT, significantly outperform other categories. PubMedBERT achieved the highest scores across all metrics. This superior performance underscores the benefits of domain-specific pre-training, which enhances model understanding and contextual interpretation, for the relation extraction task, particularly in complex fields like biomedicine. These models are highly valuable for relation extraction tasks requiring precise and reliable interpretations due to their nuanced understanding of language.

The reason for PubMedBERT’s superior performance may be domain-specific pre-training using contextual embeddings. Compared to clinicalBERT, PubMedBERT is specifically pre-trained on the biomedical literature. This gives an inherent advantage to PubMedBERT in understanding and processing biomedical texts. SciBERT, on the other hand, is trained on a broader range of scientific texts from various disciplines, which might dilute its effectiveness in the biomedical domain, potentially leading to less accurate representations of biomedical-specific terms like mRNA and miRNA in the vocabulary. This domain-specific pre-training allows PubMedBERT to capture the nuances and specialized terminology frequently used in the biomedical literature, which other models might miss. Traditional ML models like SVM and LR rely heavily on manually engineered features. In the complex domain of biomedical texts, capturing the necessary features to identify relations accurately is challenging and often insufficient compared to deep learning models that learn the features automatically. These models typically use bag-of-words or simple n-gram approaches, which do not capture the context of words as effectively as transformer-based models. This limitation hinders their ability to understand and extract relations that depend on the broader context within sentences. The performance of these models tends to peak around 0.65 because they reach the limits of what can be achieved with linear separability and the engineered features. Without the ability to capture complex patterns and dependencies in the text, their effectiveness is constrained.

LLMs such as variants of GPT and Claude show mixed results, with some configurations of GPT models scoring lower, possibly reflecting challenges in domain-specific applications without extensive fine-tuning. These results are, in turn, aligned with the other reported results for relation extraction tasks in biomedical informatics. However, Llama models demonstrate exceptional recall, suggesting their potential utility in the relation extraction applications. Llama seems to capture as many relevant instances as possible. This feature could be particularly beneficial in preliminary data exploration phases or broad information retrieval tasks. The further fine-tuning of open-sourced LLaMA models could have great potential applications in relation extraction tasks.

Overall, the observed variances in model performances suggest that the choice of model in biomedical relation extraction must be informed by the specific requirements. These include considerations of computational resources, the need for precision or recall, and the complexity of the data. While transformer-based models show great promise, their computational demands and extensive data requirements might limit their feasibility for certain other IE applications. The ongoing advancements in LLMs indicate a promising future where the models might soon close the performance gap through more refined training techniques such as fine-tuning, retrieval-augmented generation (RAG), and improved architectures. Continued research and development are crucial in leveraging these insights, potentially leading to the development of more robust, efficient, and accurate models across various domains.

Limitations and Future Enhancement: One of the major limitations of the study is model generalizability and transferability. The relation extraction models trained on specific datasets may struggle to perform well on unseen data, particularly if the training data do not sufficiently cover the variability found in real-world applications. The specificity of miRNA–mRNA interactions related to five diseases may limit the ability of the models to generalize across different sub-domains of biomedicine. Since the study focuses specifically on selected diseases, other gold-standard miRNA-gene relation extraction datasets were not evaluated.

As a future enhancement of the study to address generalizability and transferability, we will diversify the training datasets. This can be achieved by incorporating miRNA–mRNA interaction data across a broader range of diseases and experimental setups beyond the five diseases under study. To further address the issue of limited generalizability, implementing rigorous cross-dataset validation techniques would be beneficial. To further improve the performance of the best performing LLM, Llama 2, we will fine tune the model weights using RAG methods and other new techniques such as QLoRA (Quantized Low-Rank Adaptation) methods.

## 5. Conclusions

We developed an NLP/LLM pipeline to extract miRNA–mRNA interactions from PubMed abstracts and genomic analysis to understand the critical role of miRNA–mRNA interactions in various biological processes and disease mechanisms. To evaluate the pipeline, we annotated the MMIC corpus and released it as open source. We further conducted a network study to understand all the diseases linked to miRNA–mRNA in the MMIC corpus. We also performed a pathway enrichment analysis to understand the significant pathways related to miRNA–mRNA interactions. The findings from pathway analysis contribute to a deeper understanding of the molecular mechanisms of diseases and offer a roadmap for future research and therapeutic development. Overall, the study demonstrates the effectiveness of integrating NLP/LLM with genomic analysis to explore the complex roles of miRNAs in disease pathogenesis, paving the way for future experimental validation and therapeutic advancements.

## Figures and Tables

**Figure 1 biomedicines-12-01535-f001:**
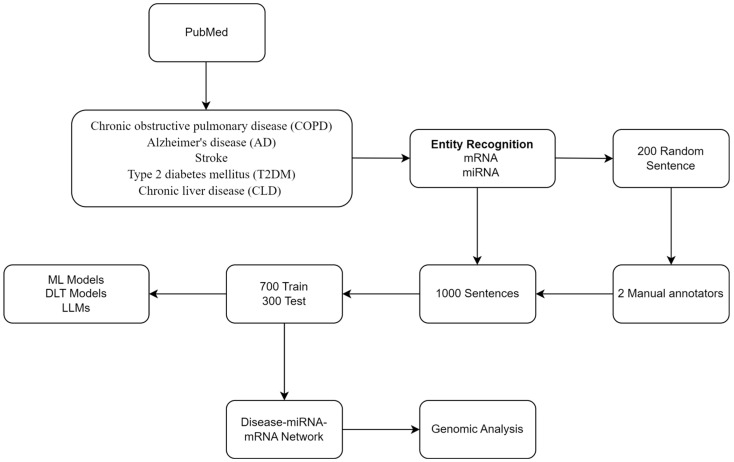
Workflow.

**Figure 2 biomedicines-12-01535-f002:**
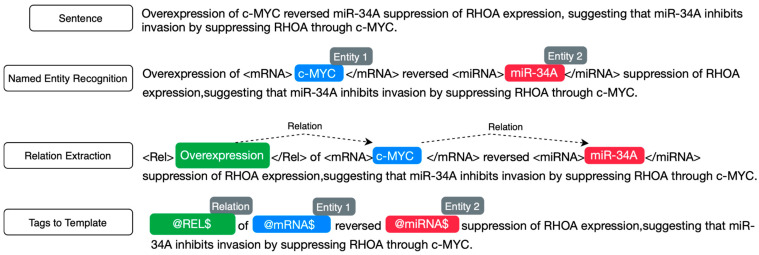
Sentence annotation pipeline.

**Figure 3 biomedicines-12-01535-f003:**
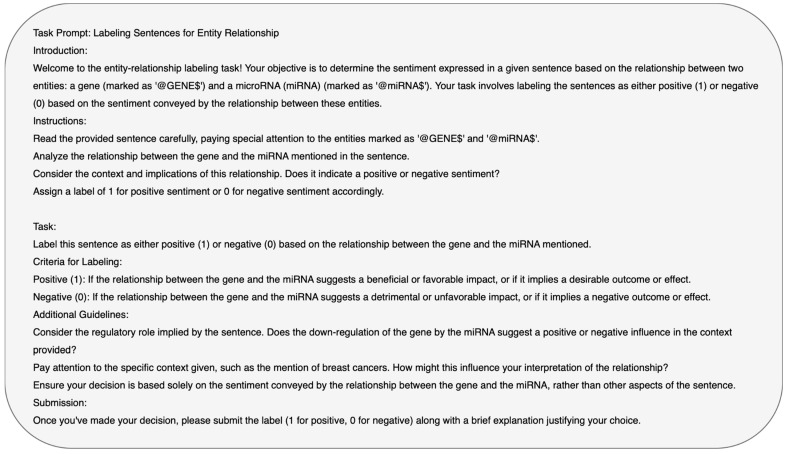
Prompt for LLM.

**Figure 4 biomedicines-12-01535-f004:**
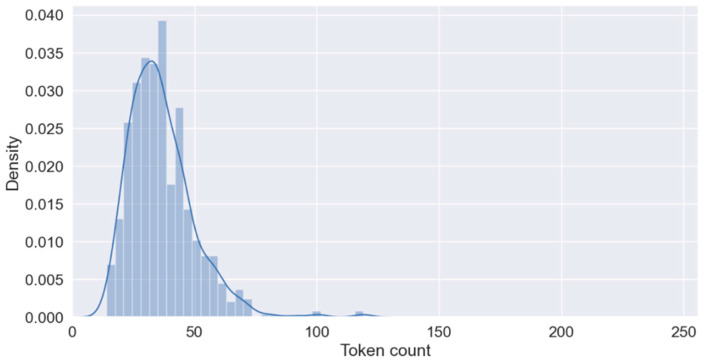
Average sentence length of training dataset.

**Figure 5 biomedicines-12-01535-f005:**
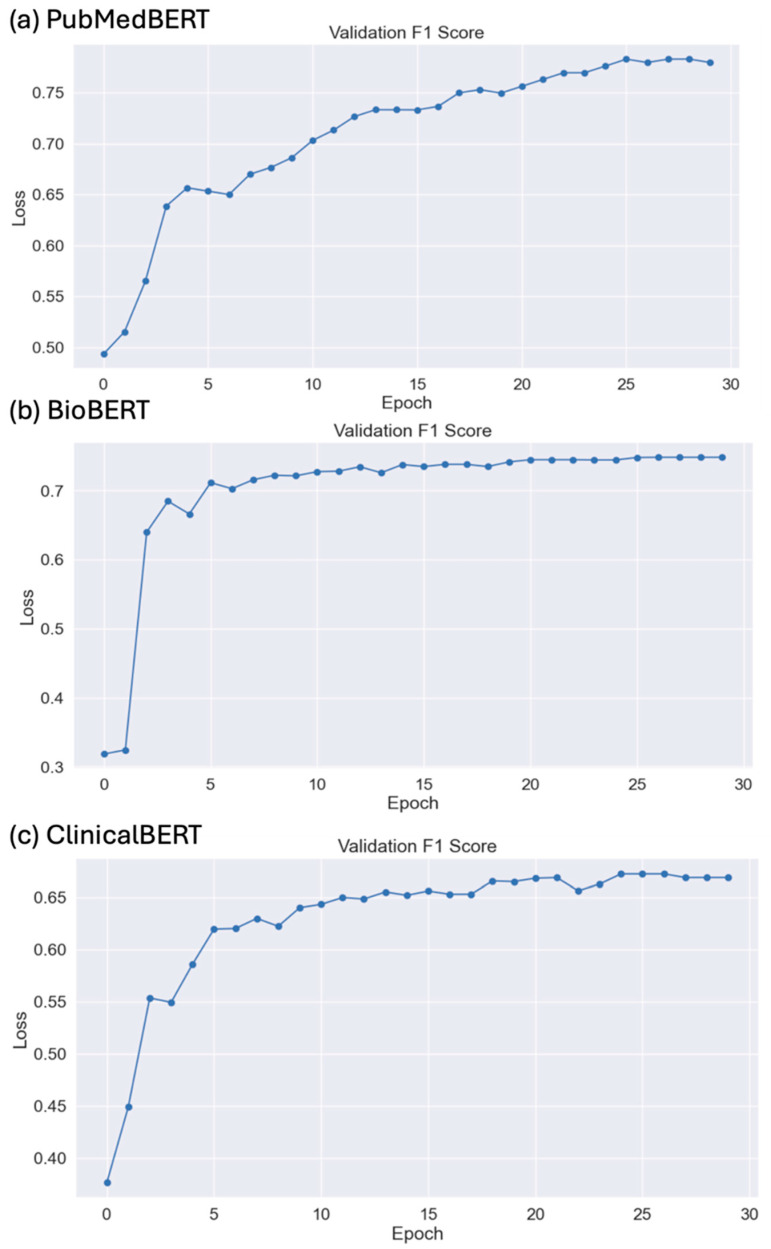
(**a**) PubMed, (**b**) BioBERT, and (**c**) ClinicalBERT F1-score on test data for 30 epochs.

**Figure 6 biomedicines-12-01535-f006:**
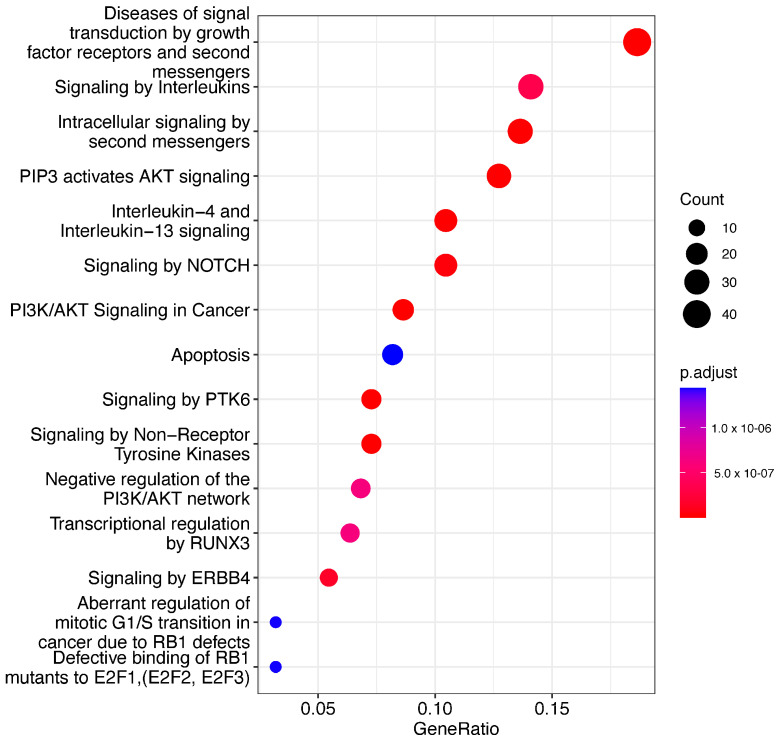
Dotplot pathway enrichment map showing the significantly over-represented pathways (q < 0.05).

**Figure 7 biomedicines-12-01535-f007:**
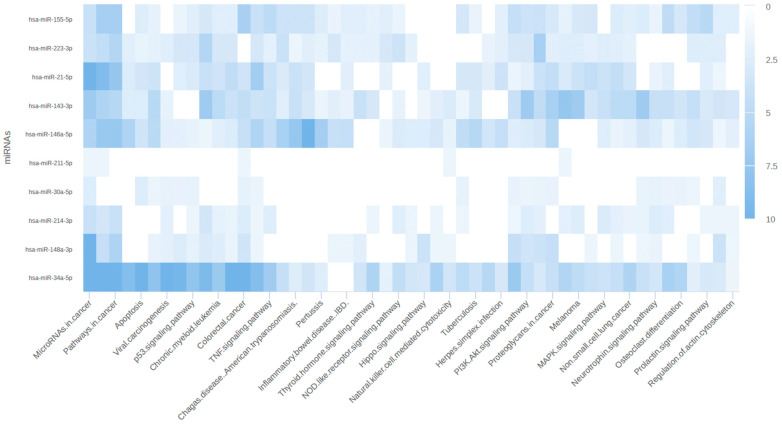
Heatmap representing pathways associated with miRNA–mRNA relations.

**Table 1 biomedicines-12-01535-t001:** Statistics of MMIC corpus.

	Positives	Negatives	Total
**Train**	354	346	700
**Test**	146	154	300
**MMIC corpus**	500	500	1000

**Table 2 biomedicines-12-01535-t002:** Performance of ML, DLT, and LLM models on MMIC corpus.

Type	Model	Experiment	Precision	Recall	F-Score
**ML**	SVM	-	0.620	0.620	0.620
	Logical regression	-	0.650	0.650	0.650
	KNN	-	0.550	0.540	0.530
	Decision tree	-	0.650	0.650	0.650
	Random forest	-	0.656	0.656	0.650
	XGBoost	-	0.670	0.670	0.660
	LightGBM	-	0.650	0.640	0.640
**DLT**	ClinicalBERT	-	0.674	0.673	0.672
	BioBERT	-	0.754	0.750	0.748
	PubMedBERT	-	** 0.783 **	0.783	** 0.783 **
**LLM**	GPT-3.5	Zero-shot	0.536	0.664	0.593
		Three-shot	0.523	0.384	0.443
	GPT-4	Zero-shot	0.484	0.103	0.170
		Three-shot	0.476	0.137	0.213
	Claude 2	Zero-shot	0.587	0.507	0.544
		Three-shot	0.584	0.452	0.510
	Llama 2	Zero-shot	0.563	0.863	0.681
		Three-shot	0.555	** 0.870 **	0.677

Underline: Best performance with each type; Bold: **Best performance among all models**.

## Data Availability

The newly created miRNA–mRNA Interaction Corpus (MMIC) and codes are available at https://github.com/balubhasuran/miRNA_mRNA-Relation-Extraction (accessed on 2 July 2024).

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
