# Peer review of "Large Language Models and Genomics for Summarizing the Role of microRNA in Regulating mRNA Expression"

_biomedicines, 2024, doi:10.3390/biomedicines12071535_

Round 1
Reviewer 1 Report
Comments and Suggestions for Authors
This manuscript describes the annotation of a miRNA-mRNA relation extraction corpus known as mmREC, which is utilized to evaluate Machine Learning (ML) models, Deep Learning-based Transformer (DLT) models, and Large Language Models (LLMs). Additionally, the authors have applied ML, DLT, and LLM models to extract miRNA-mRNA relationships from PubMed.
- The authors state in the abstract that they have established mmREC, but there is no introduction to mmREC in the main text. I am uncertain about the contents of this corpus.
- The introduction section discusses several methods for extracting miRNA-mRNA relationships from PubMed, such as RIscoper 2.0, which achieved high F-scores. However, in the results section, the authors do not compare their methods with these, leaving the readers uninformed about the advantages of the methods presented in this paper.
- The results in Section 3.2 show that the accuracy, recall, and F-scores of the methods presented in this manuscript are not very satisfactory. Overall, the performance of the LLM methods appears to be lower than that of the ML and DLT methods. Why is this the case? The authors need to analyze these results. Is the LLM approach not suitable for this corpus?
- There is an issue with the formatting of the caption for Table 2.
Comments on the Quality of English LanguageMinor editing of English language required.
Author Response
Reviewer 1:
This manuscript describes the annotation of a miRNA-mRNA relation extraction corpus known as mmREC, which is utilized to evaluate Machine Learning (ML) models, Deep Learning-based Transformer (DLT) models, and Large Language Models (LLMs). Additionally, the authors have applied ML, DLT, and LLM models to extract miRNA-mRNA relationships from PubMed.
- The authors state in the abstract that they have established mmREC, but there is no introduction to mmREC in the main text. I am uncertain about the contents of this corpus.
We appreciate the reviewer's suggestion. Now, we have renamed the corpus as miRNA-mRNA Interaction Corpus (MMIC) and used it throughout the manuscript. We have included the details about the number of positive and negative sentences, as well as the entities present, in Section 3.1 (page 8, lines 317-323). We also summarize the details in Table 1.
- The introduction section discusses several methods for extracting miRNA-mRNA relationships from PubMed, such as RIscoper 2.0, which achieved high F-scores. However, in the results section, the authors do not compare their methods with these, leaving the readers uninformed about the advantages of the methods presented in this paper.
We appreciate the reviewer's suggestion. The studies like RIscoper 2.0 focus on all types of RNA-RNA extraction and do not specifically annotate and evaluate miRNA-mRNA interactions. To avoid confusion, we removed such citations from the introduction section of the manuscript and kept only the miRNA-mRNA based studies.
Among these, two existing works, miRTex and miRSel, are released as web-based tools. The dataset used for evaluating miRTex is released as a structured text that includes PMID, miRNA, mRNA, direction, and relation type. However, the exact sentence with miRNA and mRNA from the PubMed abstract is not available. For a fair comparison with our pipeline, we need the exact dataset with annotation used by miRTex. The dataset used for evaluating miRSel is not available. Thus, evaluating our approach on the dataset used for miRSel is impossible. The dataset used for evaluating IBRel and the approach developed by Lou et al. are available as free-text without entity-level annotation for miRNA and mRNA. It is impossible to evaluate our approach on this dataset without the entity-level annotation.
We also explored the possibility to evaluate the existing approaches on our MMIC corpus. The web-interface of both miRTex and miRSel are not functioning. Though the source code is available for IBRel and the approach developed by Lou et al., it is impossible to validate their performance on MMIC corpus that includes annotations for miRNA and mRNA in the input sentences.
We have included the details in the discussion section (pages 15-16, lines 420-435).
- The results in Section 3.2 show that the accuracy, recall, and F-scores of the methods presented in this manuscript are not very satisfactory. Overall, the performance of the LLM methods appears to be lower than that of the ML and DLT methods. Why is this the case? The authors need to analyze these results. Is the LLM approach not suitable for this corpus?
Although large language models (LLMs) are known to excel in tasks such as summarization and reasoning, there are reports indicating that LLMs underperform in information extraction (IE) tasks, particularly in biomedical domains such as relation extraction. This study investigates this issue specifically for the miRNA-mRNA relationship. Our recent study on exploring the performance of LLM on relation extraction using three standard corpora namely EU-ADR, GAD, and ChemProt validates the above statement. Our study clearly shows the lower performance of LLM when compared to BioBERT and PubMedBERT. We have included the details and cited our recent study in page 16, lines 437-443.
Our recent study supporting the lower performance of LLMs on relation extraction using standard corpora is given below:
Zhang J, Wibert M, Zhou H, Peng X, Chen Q, Keloth VK, Hu Y, Zhang R, Xu H, Raja K. A Study of Biomedical Relation Extraction Using GPT Models. AMIA Jt Summits Transl Sci Proc. 2024 May 31;2024:391-400. PMID: 38827097; PMCID: PMC11141827.
- There is an issue with the formatting of the caption for Table 2.
We thank the reviewer for the comment. Table 2 caption is formatted accordingly.
Minor editing of English language required.
We appreciate the reviewer's suggestion. The entire manuscript has been thoroughly revised and updated.
Reviewer 2 Report
Comments and Suggestions for Authors
This paper evaluated the performance of a variety of machine learning (ML) models, deep learning-based transformer (DLT) models, and large language models (LLMs) to extract the miRNA-mRNA relations mentioned in PubMed. They also used the genomics approaches for validating the extracted miRNA-mRNA relations. To improve the paper:
1. It was mentioned that all the ML Models were trained in the default setting. ML models often have numerous hyperparameters that can significantly influence their performance. Default settings may not be the best fit for this study because the models might not achieve their full potential in terms of accuracy, precision, recall, or F1-score.
2. It was mentioned in line 305-306 that “The pathways with significant p-values of < 0.05, associated with mRNA in KEGG and Reactome Pathway databases were retrieved.” Can authors explain what is the hypothesis test for the p-value in the Pathway Enrichment Analysis?
3. Please add more details on why q-values (q < 0.05) instead of p-values (p < 0.05) in pathway enrichment analysis was used (Figure 7)
4. Please provide a more detailed analysis of why certain models performed better or worse in the discussion. And discuss what factors might have influenced these results. For example, explain why PubMedBERT outperformed other models and why ML models like SVM and Logistic regression show moderate effectiveness, with performance peaking around 0.65.
5. This study utilized miRPathDB for pathway analysis and visualized the results as a heatmap in Figure 8. However, a more detailed interpretation of the heatmap is lacking.
6. The authors have cited 8 works involving Bhasuran, B. and collaborators. Please justify that each self-citation is directly relevant to the content of the current paper.
7. Repeated Descriptions of Models. Please merge repeated descriptions of into a single, detailed description in the methodology section.
8. The discussion and conclusion sections of the paper appear to be somewhat repetitive and redundant in terms of the Model Performance. Please discuss the performance of different models in detail in the discussion section and provide a concise summary in the conclusion.
Author Response
Reviewer 2:
This paper evaluated the performance of a variety of machine learning (ML) models, deep learning-based transformer (DLT) models, and large language models (LLMs) to extract the miRNA-mRNA relations mentioned in PubMed. They also used the genomics approaches for validating the extracted miRNA-mRNA relations. To improve the paper:
- It was mentioned that all the ML Models were trained in the default setting. ML models often have numerous hyperparameters that can significantly influence their performance. Default settings may not be the best fit for this study because the models might not achieve their full potential in terms of accuracy, precision, recall, or F1-score.
We appreciate the reviewer's suggestion. The aim of this study is to evaluate the performance of the LLM in extracting miRNA-mRNA relations. We included ML models as baseline and kept the default settings for performance evaluation. Many existing works have used the default settings for ML models for relation extraction.
- It was mentioned in line 305-306 that “The pathways with significant p-values of < 0.05, associated with mRNA in KEGG and Reactome Pathway databases were retrieved.” Can authors explain what is the hypothesis test for the p-value in the Pathway Enrichment Analysis?
We appreciate the reviewer’s suggestion. We have included the details of the hypothesis test in the methods section (pages 8-9, lines 293-311).
- Please add more details on why q-values (q < 0.05) instead of p-values (p < 0.05) in pathway enrichment analysis was used (Figure 7)
In Figure 7, we used q-values (q < 0.05) instead of p-values (p < 0.05) to ensure that the pathways identified as significantly enriched are truly significant, reducing the likelihood of false positives and increasing the robustness of the results. The details of the q-values are included in the methods section (page 8, lines 306-312) results section (pages 13-14, lines 381-390).
- Please provide a more detailed analysis of why certain models performed better or worse in the discussion. And discuss what factors might have influenced these results. For example, explain why PubMedBERT outperformed other models and why ML models like SVM and Logistic regression show moderate effectiveness, with performance peaking around 0.65.
We appreciate the reviewer's suggestion. Details about model performance have been included in the discussion section (pages 15-16, lines 420-435).
- This study utilized miRPathDB for pathway analysis and visualized the results as a heatmap in Figure 8. However, a more detailed interpretation of the heatmap is lacking.
We appreciate the reviewer's suggestion. The heatmap shows the number of pathways associated with the miRNA-mRNA interactions. The color variation represents the number. We have included additional text to the manuscript (page 9, lines 312-314).
- The authors have cited 8 works involving Bhasuran, B. and collaborators. Please justify that each self-citation is directly relevant to the content of the current paper.
We appreciate the reviewer's comment. All citations pertain to relation extraction tasks, particularly those using the standard gold corpus. All citations are relevant to the current study.
- Repeated Descriptions of Models. Please merge repeated descriptions of into a single, detailed description in the methodology section.
We appreciate the reviewer's suggestion. We have updated the manuscript accordingly.
- The discussion and conclusion sections of the paper appear to be somewhat repetitive and redundant in terms of the Model Performance. Please discuss the performance of different models in detail in the discussion section and provide a concise summary in the conclusion.
We appreciate the reviewer's suggestion. We have updated the discussion and conclusion sections as per the suggestion (page 18, lines 527-538).
Round 2
Reviewer 2 Report
Comments and Suggestions for Authors
All comments have been addressed